# Are elevated plasma fibrinogen associated with lung function? An 8-year follow-up of the ELSA study

Camila Thais Adam[1], Ione Jayce Ceola Schneider[1]*, Danielle Soares Rocha Vieira[1], Tauana Prestes Schmidt[1], Fernando Cesar Wehrmeister[2], Cesar de Oliveira[3]

**1** Post-graduate Program in Rehabilitation Sciences, Federal University of Santa Catarina, Araranguá, Santa Catarina, Brazil, **2** Department of Epidemiology, Federal University of Pelotas, Pelotas, Rio Grande do Sul, Brazil, **3** Department of Epidemiology & Public Health, University College London, London, United Kingdom

* ione.schneider@ufsc.br

## Abstract

### Background

Fibrinogen is an important biomarker of inflammation, but findings from longitudinal studies that correlated fibrinogen with lung function in older adults are inconsistent.

### Aim

To investigate the relationship between fibrinogen plasma levels and lung function impairment later in life.

### Methods

Longitudinal analysis of 2,150 participants of the English Longitudinal Study of Ageing (ELSA) aged 50 years and older. Associations between changes in plasma fibrinogen between waves 2 (2004–05) and 4 (2008–09) and lung function in wave 6 (2012–13) were performed using multiple linear regression adjusted by potential confounders.

### Results

Regarding the fibrinogen profile, 18.5% of the participants presented higher levels in both waves. In the adjusted models, the maintenance of high fibrinogen levels was associated with a significant reduction of lung function only for men. $FEV_1$ showed a reduction of 0.17L, FVC of 0.22L, and the percentages predicted were 5.16% for $FEV_1$ and 6.21% for FVC compared to those that maintained normal levels of fibrinogen.

### Discussion

To the best of our knowledge, this was the first study investigating the relationship between changes in fibrinogen levels over a long follow-up period and lung function in older adults without pre-existing chronic diseases. ELSA has information on critical demographic and clinical parameters, which allowed to adjust for potential confounding factors.

**Data Availability Statement:** The English Longitudinal Study of Ageing data are available to the scientific community from the UK Data Service for researchers who meet the criteria for access to

confidential data, under conditions of the End User License http://ukdataservice.ac.uk/media/455131/cd137-enduserlicence.pdf. The data can be accessed from: https://beta.ukdataservice.ac.uk/datacatalogue/series/series?id=200011#!/access-data. Contact with the UK Data Service regarding access to the English Longitudinal Study of Ageing can be made through the website https://www.ukdataservice.ac.uk/about-us/contact, by phone +44 (0)1206 872143 or by email at help@ukdataservice.ac.uk.

**Funding:** We declare that CTA received a scholarship from the Coordination for the Improvement of Higher Education Personnel (CAPES), (Financial Code 001). The National Research Council (CNPq) supports the FCW.

**Competing interests:** The authors have delcared that no competing interests exist.

## Conclusion

It was found that the persistence of high levels of plasma fibrinogen in older English men, but not women, is associated with lung function decline. Therefore, plasma fibrinogen showed to be an important biomarker of pulmonary dysfunction in this population.

## 1. Introduction

Fibrinogen is a soluble protein involved in the blood clotting mechanism [1], and it is linked to pathological processes of lung diseases [2,3]. Inflammatory stimuli induce the lung's alveolar epithelial cells to synthesize and secrete fibrinogen, demonstrating the importance of this biomarker in the inflammatory process [4,5]. In healthy lungs, low levels of fibrinogen are expected [6]. Experimental studies in animal models suggest that fibrinogen may be an essential effector molecule in airway disease [7,8]. Fibrinogen in the lungs can inactivate pulmonary surfactant, which causes increased surface-tension relationships, promote the expression of molecules that induces airway fibrosis, and activate plasminogen activator inhibitor type-1, leading to excess fibrin deposition in the airways and airway narrowing [9].

Fibrinogen plasma levels have been associated with reduced lung function in cross-sectional [10,11] and few longitudinal [12–14] analyses. Longitudinal studies were performed in the younger population [12,14] or individuals with pre-existing underlying diseases [13]. Other studies report that people with respiratory diseases had significantly higher levels of fibrinogen compared to controls [15–22]. However, the relationship between lung function and changes in systemic fibrinogen levels over time in community-dwelling older adults is not clear. Therefore, this study aimed to investigate the relationship between fibrinogen plasma levels and lung function changes later in life. We used eight years of data from a nationally representative sample of older English men and women. We hypothesized that there is a relationship between the change in plasma fibrinogen levels and lung function reduction overtime in an aging cohort.

## 2. Methods

### 2.1 Study population

Longitudinal study using data collected from the English Longitudinal Study of Ageing (ELSA), an ongoing panel study of community-dwelling men and women living in England aged 50 years and older that commenced in 2002. The ELSA sample was drawn from participants that had previously participated in the Health Survey for England. After baseline, follow-up interviews occur biannually and health examinations every four years. A detailed description of the study can be found elsewhere [23]. The National Research Ethics Service (London Multicentre Research Ethics Committee, MREC/01/2/91) approved the ELSA study. All participants gave written informed consent. Participants who had fibrinogen data from waves 2 and 4, and lung function tests at wave 6 were included in the study. The data collection process and methodology are fully described elsewhere [23]. This study followed the Strengthening the Reporting of Observational Studies in Epidemiology (STROBE) reporting guideline.

### 2.2 Variables

**2.2.1 Fibrinogen.** Blood samples were taken from willing ELSA core members, except those who had clotting or bleeding disorders, had ever had fits or convulsions, were not willing

to give their consent in writing, or were currently on anticoagulant drugs. Respondents over 80 years, those known to be diabetic and on treatment, those with a clotting or bleeding disorder or on anticoagulant drugs, those who had ever had fits, those who seemed frail, or respondents whose health was a cause for concern for the nurse were not asked to fast. Fibrinogen analysis was carried out by the Department of Haematology at the Royal Victoria Infirmary, The Newcastle Upon Tyne NHS Foundation Trust. The coagulation analyser MDA 180 (Organon Teknika, Durham, USA) was used with a modification of the Clauss thrombin clotting method by one citrate blue tube [23]. The measurement of fibrinogen was done in the ACL TOP CTS analyser. The Fibrinogen assay on the IL TOP is a clotting assay. Analyser prepares a 1 in 10 dilution of plasma in factor diluent, by aspirating 17μl of plasma and mixing it with 153μl of factor diluent. 100μl of the diluted plasma was dispensed into a reaction cuvette and incubated for 60 seconds at 37˚C. 50μl of Fibrinogen-C XL reagent was added, and the reaction monitored at 405nm for up to 120 seconds. The analyser monitors the change in light transmission caused by the conversion of soluble fibrinogen in plasma to cross-linked insoluble fibrin, and the clotting time threshold is determined to be 37% of the total change. The clotting time is directly related to the concentration of fibrinogen in the plasma. This time is converted to concentration in g/L by the automatic use of a calibration curve. For the analyses, we created a variable called fibrinogen profile which considered plasma fibrinogen levels between waves 2 and 4 (normal levels in waves 2 and 4, normal in waves 2 and altered in wave 4, altered in wave 2 and normal in waves 4; altered in waves 2 and 4). Upper tertile values of fibrinogen were considered altered ($\geq$3.8 g/L for wave 2 and $\geq$3.9 g/L for wave 4).

**2.2.2 Lung function.** A spirometer (NDD Easy On-PC) was used to evaluate forced expiratory volume in the first second ($FEV_1$) and forced vital capacity (FVC). At least three acceptable measurements were obtained, and the largest $FEV_1$ and FVC values were considered. Respondents were excluded if they had had abdominal or chest surgery in the preceding three weeks, had been admitted to hospital with a heart complaint in the previous six weeks, had had eye surgery in the prior four weeks, had a tracheotomy, or were pregnant. Further, the tests were not done if the ambient temperature was less than 15˚C or more than 35˚C, as this affects the accuracy of the readings. $FEV_1$ and FVC were expressed in absolute values (in liters), and percentage of predicted [24] were used for the analyses. All the lung function values refer to wave 6.

**2.2.3 Covariates.** Sex (female, male), household wealth (in quintiles), physical activity level (sedentary, mild, moderate, high), smoking status (non-smoker, former smoker, current smoker), physical activity (sedentary, mild, moderate, high), presence of one or more chronic diseases (osteoarthritis, cancer, osteoporosis, dementia, depressive symptoms, Alzheimer's and Parkinson's diseases), presence of one or more cardiocirculatory diseases (high blood pressure, angina, heart attack, congestive heart failure, heart murmur, and abnormal heart rhythm) and the presence of lung diseases (chronic obstructive pulmonary disease and asthma).

## 2.3 Statistical analysis

The variables of the study were plasma fibrinogen levels between waves 2 and 4 (normal levels in waves 2 and 4, normal in wave 2 and changed in wave 4, changed in wave 2 and normal in wave 4; changed in waves 2 and 4). The continuous variables were age (years), body mass index (kg/m$^2$), height (m), $FEV_1$ (litres and percentage of predicted) and FVC (litters and percentage of predicted) [23].

Categorical variables were analysed with frequencies and their respective 95% confidence intervals (CI). Continuous variables were described as mean and standard deviation. Associations between fibrinogen profile and $FEV_1$ (outcome) were performed using multiple linear

regression adjusted by age, wealth, physical exercise, current smoking status, chronic diseases, lung diseases, body mass index, and height. Crude and adjusted regression coefficients and 95%CI were estimated. Cardiocirculatory diseases were not included in the adjusted model because of the collinearity with the fibrinogen levels. Analyses were sex-stratified. The following post-estimation tests were performed: Shapiro-Wilk and Shapiro-Francia test for normality and Breuschof-Pagan and Cook-Weisberg test for heteroskedasticity. We also performed the command *margins* to calculate the predictions of the marginal effects of the variable fibrinogen. This information is presented in plots. These procedures were repeated with absolute values of FVC and percentage of predicted $FEV_1$ and FVC. Analyses were performed in Stata SE version 14.0.

## 3. Results

The analytical sample included 2,150 participants from the ELSA cohort who had fibrinogen data in waves 2 and 4 and lung function data in wave 6. Fig 1 presents the flowchart with the selection process between waves 2 and 6. The follow-up period was 8 years.

Table 1 shows sociodemographic and health variables frequency distribution (all sample and sex-stratified). The majority of participants were female (55.9%), belonging to the lowest wealth quintile (24.0%), with moderate physical activity level (51.9%), classified as ex-smokers (54.2%), with chronic diseases (56.6%) and no pulmonary diseases (80.2%). Regarding the fibrinogen profile, 55.4% of the participants remained within the normal levels in waves 2 and 4, and 18.5% showed higher levels in both waves. The main differences found between men and women were related to smoking, physical activity, and the presence of chronic diseases. Women had lower $FEV_1$ and FVC values than men. The $FEV_1$ and FVC percentage of predicted values were similar for men and women.

The associations between lung function parameters and fibrinogen categories are presented in Table 2. For men, in the crude analysis, the fibrinogen profile was associated with all pulmonary function measurements for all categories, except for FVC percentage of predicted in the changed/normal category. In the adjusted analysis, only the maintenance of high levels of fibrinogen was associated with a significant reduction of lung function measures. $FEV_1$

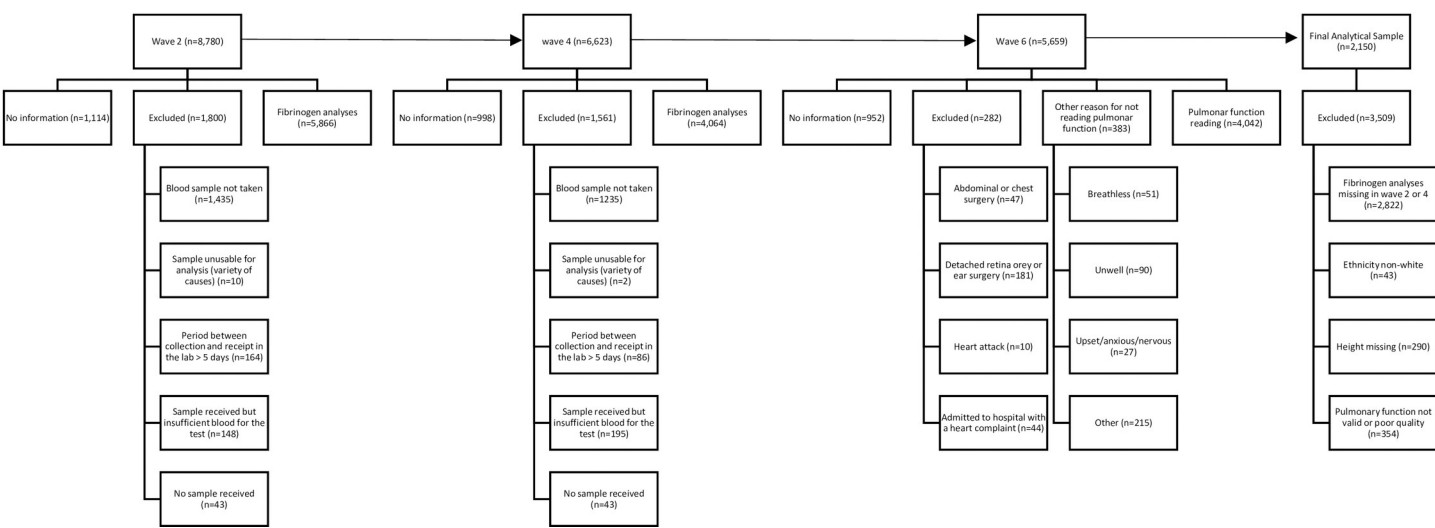

**Fig 1. Flowchart describing the sample selection from waves 2 (2004/5), 4 (2008/9) and 6 (2012/13), The English Longitudinal Study of Ageing, England, 2004 to 2013.**

**Table 1. Descriptive analyses of participants from ELSA who presented complete data for fibrinogen in wave 2 (2004–05) and 4 (2008–09) and lung function in wave 6 (2012–13), The English Longitudinal Study of Ageing, England, 2004 to 2013.**

| Variables | | General | | Male | | Female | |
|---|---|---|---|---|---|---|---|
| | | n | % (95%CI) | n | % (95%CI) | n | % (95%CI) |
| Sex | | | | | | | |
| | Male | 948 | 44.1 (42.0;46.2) | | | | |
| | Female | 1,202 | 55.9 (53.8;58.0) | | | | |
| Wealth | | | | | | | |
| | Lowest quintile (poorest) | 264 | 12.5 (11.1;13.9) | 104 | 11.1 (9.3;13.3) | 160 | 13,5 (11.6;15.5) |
| | 2nd quintile | 343 | 16.2 (14.7;17.8) | 136 | 14.6 (12.5;17.0) | 207 | 17.4 (15.4;19.7) |
| | 3rd quintile | 497 | 23.4 (21.7;25.3) | 218 | 23.4 (20.8;26.2) | 279 | 23.5 (21.2;26.0) |
| | 4th quintile | 507 | 23.9 (22.1;25.8) | 239 | 25.6 (22.9;28.5) | 268 | 22.6 (20.3;25.0) |
| | Highest quintile (richest) | 510 | 24.0 (22.3;25.9) | 236 | 25.3 (22.6;28.2) | 274 | 23.1 (20.8;25.5) |
| Smoking status | | | | | | | |
| | Non-smoker | 795 | 37.0 (34.9;39.5) | 265 | 27.9 (25.2;30.9) | 530 | 44.1 (41.3;46.9) |
| | Former smoker | 1,166 | 54.2 (52.1;56.3) | 595 | 62.9 (59.7;65.9) | 570 | 47.4 (44.6;50.2) |
| | Current smoker | 189 | 8.8 (7.7;10.1) | 87 | 9.2 (7.5;11.2) | 102 | 8.5 (7.3;10.2) |
| Physical exercise | | | | | | | |
| | Sedentary | 70 | 3.3 (2.6;4.1) | 28 | 3.0 (2.0;4.2) | 42 | 3.5 (2.6;4.7) |
| | Low | 296 | 13.8 (12.4;15.3) | 93 | 9.8 (8.1;11.9) | 203 | 16.9 (14.9;19.1) |
| | Moderate | 1,116 | 51.9 (49.8;54.0) | 484 | 51.1 (47.9;54.2) | 632 | 52.6 (49.7;55.4) |
| | High | 668 | 31.1 (29.1;33.1) | 343 | 36.2 (33.2;39.3) | 325 | 27.0 (24.6;29.6) |
| Chronic diseases | | | | | | | |
| | No | 930 | 43.4 (41.3;45.5) | 509 | 53.8 (50.6;57.0) | 421 | 35.1 (32.5;37.9) |
| | Yes | 1,214 | 56.6 (54.5;58.8) | 437 | 46.2 (43.0;49.4) | 777 | 65.8 (62.1;67.5) |
| Lung diseases | | | | | | | |
| | No | 1,725 | 80.2 (78.5;81.9) | 781 | 82.4 (78.8;84,7) | 944 | 78.5 (76.1;80.8) |
| | Yes | 425 | 19.8 (18.1;21.5) | 167 | 17.6 (15.3;20.2) | 258 | 21.5 (19.2;23.9) |
| Fibrinogen profile | | | | | | | |
| | Normal | 1,191 | 55.4 (53.3;57.5) | 575 | 60.6 (57.5;63.7) | 616 | 51.2 (48.4;54.1) |
| | Normal/changed | 296 | 13.8 (12.4;15.3) | 121 | 12.8 (10.8;15.0) | 175 | 14.6 (12.7;16.7) |
| | Changed/normal | 265 | 12.3 (11.0;13.8) | 105 | 11.1 (9.2;13.2) | 160 | 13.3 (11.5;15.4) |
| | Changed | 398 | 18.5 (16.9;20.2) | 147 | 15.5 (13.3;18.0) | 251 | 20.9 (18.7;23.3) |
| | | n | Mean (SD) | n | Mean (SD) | n | Mean (SD) |
| Age (years) | | 2,150 | 70.5 (7.1) | 948 | 70.5 (7.1) | 1,202 | 70.5 (7.1) |
| Height (cm) | | 2,143 | 165.1 (9.4) | 941 | 172.8 (6.6) | 1,202 | 159.2 (6.4) |
| Body mass index (kg/m$^2$) | | 2,104 | 27.9 (4.6) | 908 | 27.6 (3.6) | 1,196 | 28.1 (5.3) |
| FEV$_1$ (Liters) | | 2,103 | 2.3 (0.7) | 928 | 2.7 (0.7) | 1,175 | 1.9 (0.4) |
| FEV$_1$ percentage of predicted | | 1,985 | 92.1 (16.6) | 815 | 90.7 (15.6) | 1,170 | 93.1 (17.2) |
| FVC (Liters) | | 2,129 | 3.2 (0.9) | 940 | 3.9 (0.8) | 1,189 | 2.7 (0.6) |
| FVC percentage of predicted | | 2,082 | 98.5 (15.7) | 915 | 98.2 (15.7) | 1,167 | 98.7 (15.7) |

FEV$_1$: Forced expiratory volume in the first second; FVC: Forced vital capacity.

95%CI: 95% confidence interval; SD: Standard deviation.

showed a reduction of 0.17L, FVC of 0.22L, and the percentage of predicted was 5.16 percentage points for FEV$_1$ and 6.21 percentage points for FVC compared to those that maintained normal levels of fibrinogen. Fig 2 shows the linear predictions of the mean values for each category for men.

**Table 2. Regression of FEV₁ (L), FVC (L), FEV₁ percentage of predicted and FVC percentage of predicted, by gender, The English Longitudinal Study of Ageing, England, 2004 to 2013.**

| Lung function | | n | Fibrinogen profile | | | | | |
|---|---|---|---|---|---|---|---|---|
| | | | Normal/changed | | Changed/normal | | Changed | |
| | | | Coefficient (95%CI) | p-value | Coefficient (95%CI) | p-value | Coefficient (95%CI) | p-value |
| **Men** | | | | | | | | |
| FEV₁ (L) | | | | | | | | |
| | Crude | 928 | **-0.21 (-0.33;-0.08)** | **0.002** | **-0.24 (-0.38;-0.10)** | **0.001** | **-0.42 (-0.54;-0.30)** | **<0.001** |
| | Adjusted* | 866 | -0.06 (-0.17;0.04) | 0.248 | -0.05 (-0.16;0.06) | 0.391 | **-0.17 (-0.27;-0.07)** | **0.001** |
| Percentual predicted FEV₁ | | | | | | | | |
| | Crude | 815 | **-3.40 (-6.66;-0.13)** | 0.041 | -2.08 (-5.47;1.30) | 0.228 | **-7.93 (-11.0;-4.88)** | **<0.001** |
| | Adjusted* | 758 | -1.86 (-5.05;1.33) | 0.253 | -0.11 (-3.46;3.23) | 0.948 | **-5.16 (-8.31;-2.00)** | **0.001** |
| FVC (L) | | | | | | | | |
| | Crude | 940 | **-0.25 (-0.41;-0.09)** | **0.002** | **-0.21 (-0.38;-0.04)** | **0.015** | **-0.50 (-0.65;-0.35)** | **<0.001** |
| | Adjusted* | 878 | -0.07 (-0.20;0.05) | 0.259 | 0.02 (-0.11;0.16) | 0.753 | **-0.22 (-0.34;-0.09)** | **0.001** |
| Percentual predicted FVC | | | | | | | | |
| | Crude | 915 | **-3.93 (-7.06;-0.81)** | 0.014 | -0.65 (-3.89;2.58) | 0.691 | **-8.39 (-11.22;-5.56)** | **<0.001** |
| | Adjusted* | 855 | -3.10 (-6.22;0.02) | 0.052 | 1.12 (-2.14;4.39) | 0.499 | **-6.21 (-9.20;-3.22)** | **<0.001** |
| **Women** | | | | | | | | |
| FEV₁ (L) | | | | | | | | |
| | Crude | 1,175 | -0.07 (-0.15;0.01) | 0.068 | -0.03 (-0.11;0.04) | 0.380 | **-0.16 (-0.22;-0.09)** | **<0.001** |
| | Adjusted* | 1,151 | -0.01 (-0.06;0.06) | 0.934 | 0.02 (-0.04;0.09) | 0.415 | -0.02 (-0.08;0.03) | 0.406 |
| Percentual predicted FEV₁ | | | | | | | | |
| | Crude | 1,170 | -0.45 (-3.32;2.47) | 0.764 | 0.30 (-2.73;3.33) | 0.764 | **-2.96 (-5.45;-0.39)** | **0.024** |
| | Adjusted* | 1,147 | 0.93 (-1.82;3.71) | 0.505 | 1.46 (-1.42;4.33) | 0.321 | -0.28 (-2.84;2.28) | 0.832 |
| FVC (L) | | | | | | | | |
| | Crude | 1,189 | -0.11 (-0.20;-0.01) | 0.031 | -0.08 (-0,18;0.02) | 0.109 | **-0.22 (-0.31;-0.14)** | **<0.001** |
| | Adjusted* | 1,165 | -0.01 (-0.09;0.06) | 0.695 | 0.01 (-0.07;0.08) | 0.856 | -0.02 (-0.09;0.05) | 0.563 |
| Percentual predicted FVC | | | | | | | | |
| | Crude | 1,167 | -2.02 (-4.69;0.64) | 0.137 | -1.39 (-4.14;1.37) | 0.323 | **-4.10 (-6.43;-1,76)** | **0.001** |
| | Adjusted* | 1,143 | -0.37 (-2.93;2.20) | 0.779 | 0.21 (-2.45;2,87) | 0.875 | -0.71 (-3.06;1.65) | 0.554 |

95% CI: 95% Confidence intervals; FEV₁: Forced expiratory volume in the first second; FVC: Forced vital capacity.

*Adjusted by age in exam, wealth in quintiles, physical exercise, current smoke habits, chronic diseases, lung diseases, height and body mass index.

For women, in the crude analysis, only the maintenance of the abnormal fibrinogen profile was associated with lung function measures. However, none of the categories were associated with lung function measures in women in the adjusted analysis (Table 2). Fig 3 shows the linear predictions of the mean values for each category for women.

## 4. Discussion

Our findings showed an inverse significant relationship between lung function and fibrinogen plasma levels in older English men, but not in women. Persistent high fibrinogen plasma levels in men were significantly associated with reduced lung function measures independently of age, wealth, physical activity, smoking, chronic diseases, lung diseases, body mass index and height.

To the best of our knowledge, this is the first study to investigate changes in fibrinogen levels over a long follow-up period and their relationship with lung function later in life in community-dwelling older adults without pre-existing lung and chronic diseases like chronic obstructive pulmonary disease, asthma, and cancer. Findings from longitudinal studies that

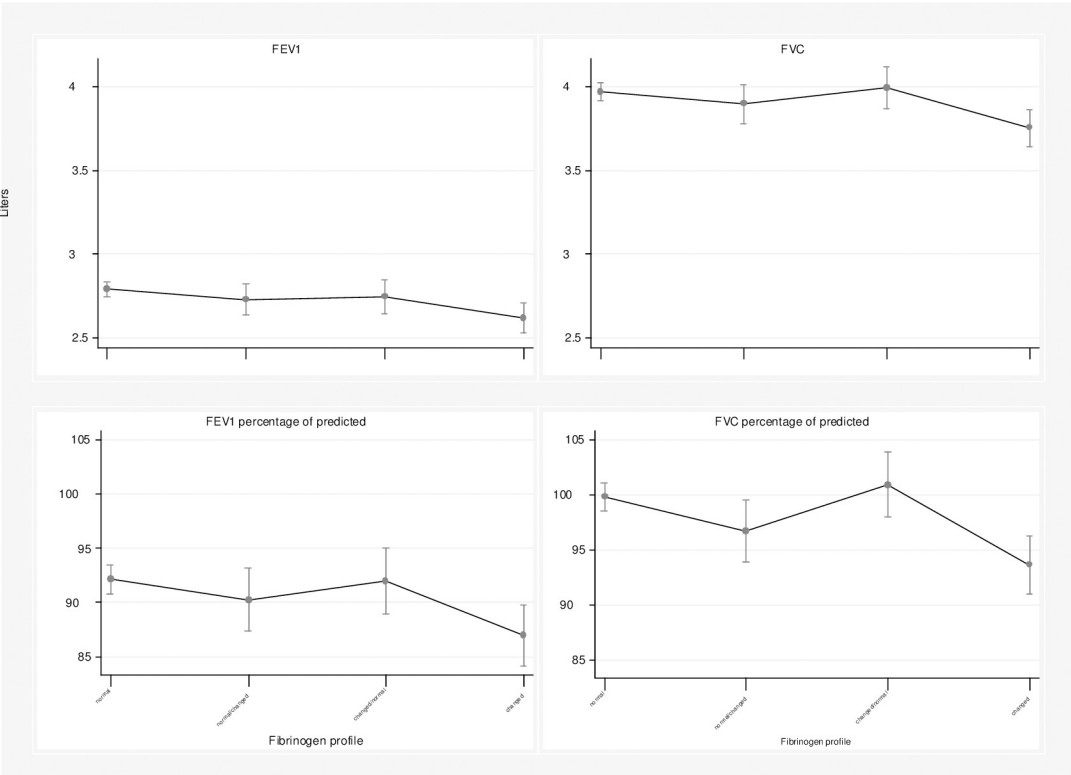

**Fig 2. Linear predictions for the FEV$_1$ (L), FVC (L), FEV$_1$ percent predicted and FVC percent predicted in men, The English Longitudinal Study of Ageing, England, 2004 to 2013.**

correlated fibrinogen with lung function in other populations are inconsistent due to different methodological designs and statistical models used, making comparability of results difficult [12–14].

Some authors emphasized that systemic inflammation may be involved in the pathogenesis of chronic lung disorders and suggest that fibrinogen may accelerate lung function decline in young adults besides predispose them to airway diseases [12,14]. However, a study carried out in New Zealand, with adults aged 32 to 38 years, showed no significant association between fibrinogen and lung function in men at age 32. In women aged 32, FEV$_1$ and FVC were inversely associated with fibrinogen. But in the longitudinal models, the authors did not find evidence that systemic inflammation predicted a decline in lung function [25]. Another study, using data from the Newcastle Thousand Families birth cohort at age 49–51 years, found significant inverse associations between both FEV$_1$ and FVC and plasma fibrinogen concentration, even after adjustments for potential confounders, including early life indicators, total pack/years of cigarette smoked and contemporaneous BMI. However, their findings did not remain significant after adjustment for contemporary percent body fat [26].

ELSA findings demonstrate that lung function decline is independent of chronic and respiratory diseases [27]. ELSA has information on important demographic and clinical parameters, which allowed us to carefully adjust potential confounding factors in our study. The most important potential confounding factor was smoking, which in itself induces systemic inflammation and is a known risk factor for pulmonary function decrease [28]. It was reassuring that, even for non-smokers and individuals with cancer or COPD, there is a relationship between the maintenance of altered fibrinogen levels and the reduction in FEV$_1$ and FVC only

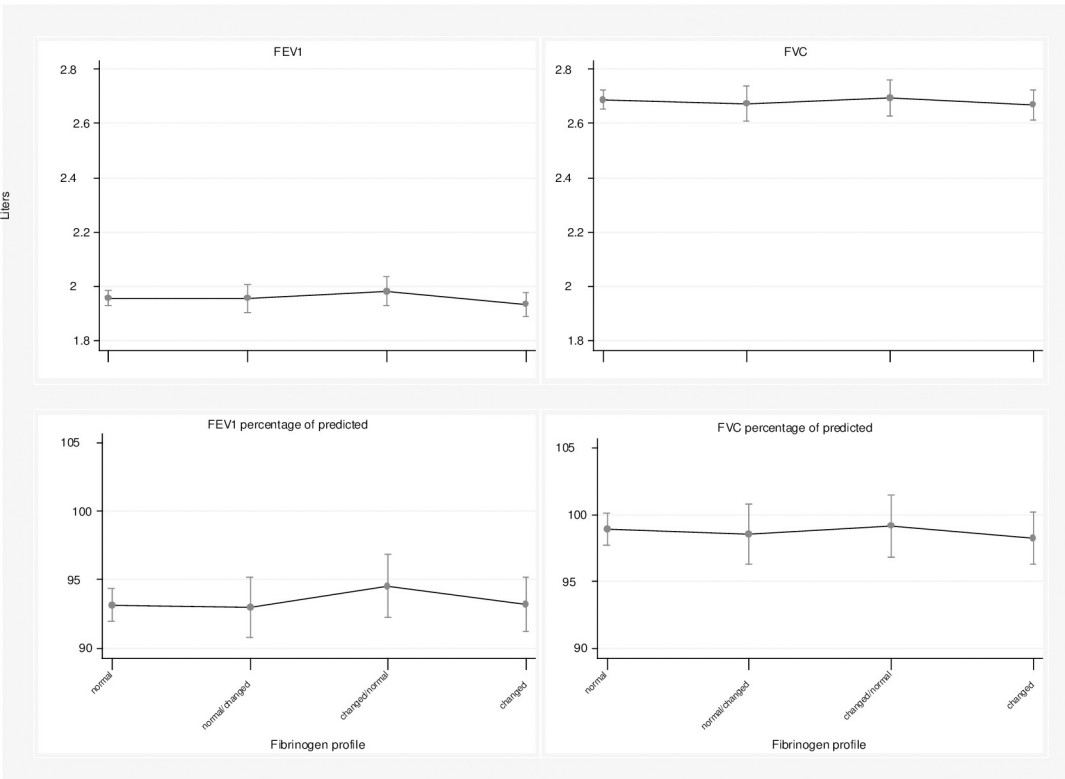

**Fig 3. Linear predictions for the FEV$_1$ (L), FVC (L), FEV$_1$ percentage of predicted and FVC percentage of predicted in women, The English Longitudinal Study of Ageing, England, 2004 to 2013.**

in men. Changes in fibrinogen levels i.e., altered to normal overtime were not associated with impaired lung function. For women, the changes in fibrinogen levels did not affect their lung function over time. So far, we cannot easily explain the sex difference in our results. It is necessary to assess whether this persists with advancing age in future ELSA cohort assessments. However, Hancox and colleagues showed that the association between inflammatory markers and lung function appears to be stronger in men than in women [25].

Data from 3,257 Japanese individuals showed that fibrinogen plasma levels were significantly associated with decreased FVC and FEV$_1$ values in men with restrictive, obstructive, and mixed ventilatory dysfunctions. However, there was no association between fibrinogen plasma levels and the reduction of pulmonary function measurements in women [11]. These results are consistent with our findings.

The association between fibrinogen levels and lung function has been investigated in people with COPD and asthma, as well as smokers. Inflammatory markers are useful in COPD to identify the risk of exacerbations and mortality [29–32]. Besides that, circulating fibrinogen was significantly increased in subjects with heart disease, hypertension, and diabetes and coexistent COPD [33]. Hyun et al.[29] demonstrated that patients with COPD with high plasma fibrinogen concentrations and low 25-OH vitamin D presented lower pulmonary function, higher COPD severity index, and higher rate of severe exacerbations over 24 months.

In individuals with asthma and chronic cough, findings have shown that levels of plasma fibrinogen are higher than in those without chronic cough [34]. In smokers with symptoms but with preserved spirometry, fibrinogen levels were not associated with respiratory symptom burden, exacerbations, 6-minute walk distance, and FEV$_1$ [35]. Previous studies have

demonstrated that high fibrinogen levels increase, especially when lung disease is present, but our findings have shown that higher levels are involved in reduction of lung function even in older adults without respiratory dysfunctions, especially in men.

The present study has some strengths that should be considered. First, we used two well-validated measures i.e., fibrinogen and spirometry from a large population-based and nationally representative cohort of English adults aged 50 years and older. These measures were collected through well-standardized procedures. Both tests could provide valuable information about diagnosis and prognosis. Secondly, lung function evaluation was standardized with quality criteria [23]. Thirdly, we used the reference prediction equations for lung function, which have been validated for most countries [24]. This study also has some limitations that need to be recognized. Survival bias could be a potential limitation. Besides, the lack of cause-specific mortality data of individuals with lung impairment information during the follow-up period. Another potential limitation of this study is the unavailability of information about the hepatic function or chronic hepatic disease. The liver plays a key role in the blood coagulation process because it is the site of synthesis of all clotting factors and their inhibitors. Liver damage is commonly associated with impairment of haemostasis. The activation of blood coagulation process is associated with decreased lung function, and that systemic inflammation may contribute to this relation [36].

## 5. Conclusion

We found that persistent high levels of plasma fibrinogen in older English men, but not in women, are associated with a reduced lung function eight years later. Our findings suggest that plasma fibrinogen may be an important biomarker of pulmonary dysfunction.

## Author Contributions

**Conceptualization:** Camila Thais Adam, Ione Jayce Ceola Schneider, Fernando Cesar Wehrmeister, Cesar de Oliveira.

**Data curation:** Camila Thais Adam, Ione Jayce Ceola Schneider, Cesar de Oliveira.

**Formal analysis:** Camila Thais Adam, Ione Jayce Ceola Schneider, Fernando Cesar Wehrmeister, Cesar de Oliveira.

**Funding acquisition:** Camila Thais Adam.

**Investigation:** Camila Thais Adam, Ione Jayce Ceola Schneider.

**Methodology:** Camila Thais Adam, Ione Jayce Ceola Schneider, Danielle Soares Rocha Vieira, Fernando Cesar Wehrmeister.

**Project administration:** Camila Thais Adam.

**Resources:** Camila Thais Adam.

**Software:** Camila Thais Adam, Ione Jayce Ceola Schneider, Tauana Prestes Schmidt, Fernando Cesar Wehrmeister.

**Supervision:** Camila Thais Adam, Ione Jayce Ceola Schneider, Fernando Cesar Wehrmeister, Cesar de Oliveira.

**Validation:** Camila Thais Adam, Danielle Soares Rocha Vieira.

**Visualization:** Camila Thais Adam, Danielle Soares Rocha Vieira, Tauana Prestes Schmidt, Cesar de Oliveira.

**Writing – original draft:** Camila Thais Adam, Danielle Soares Rocha Vieira.

**Writing – review & editing:** Camila Thais Adam, Ione Jayce Ceola Schneider, Danielle Soares Rocha Vieira, Tauana Prestes Schmidt, Fernando Cesar Wehrmeister, Cesar de Oliveira.

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
