## [Decision Letter · Decision Letter 0]

17 Jun 2021

PONE-D-20-39275

Are elevated plasma fibrinogen associated with lung function? An 8-year follow-up of the ELSA study

PLOS ONE

Dear Dr. Schneider,

Thank you for submitting your manuscript to PLOS ONE. After careful consideration, we feel that it has merit but does not fully meet PLOS ONE’s publication criteria as it currently stands. Therefore, we invite you to submit a revised version of the manuscript that addresses the points raised during the review process.

We look forward to receiving your revised manuscript.

Kind regards,

Emiliano Giardina

Academic Editor

PLOS ONE

Journal Requirements:

2. In the methods section please provide details regarding how house hold wealth and physical exercise was stratified for analysis.

"The English Longitudinal Study of Ageing was developed by a team of researchers based at the University College London, NatCen Social Research, the Institute for Fiscal Studies and the University of Manchester. The NatCen Social Research collected the data. The National Institute of Aging Grant R01AG017644 and a consortium of UK government departments coordinated by the Economic and Social Research Council provide the funding."

"Funders did not participate in the study design, data collection and analysis, decision

to publish or prepare the manuscript."

Reviewers' comments:

Reviewer's Responses to Questions

**Comments to the Author**

1. Is the manuscript technically sound, and do the data support the conclusions?

Reviewer #1: Yes

Reviewer #2: Partly

Reviewer #3: Partly

2. Has the statistical analysis been performed appropriately and rigorously? 

Reviewer #1: Yes

Reviewer #2: Yes

Reviewer #3: Yes

3. Have the authors made all data underlying the findings in their manuscript fully available?

Reviewer #1: Yes

Reviewer #2: Yes

Reviewer #3: Yes

4. Is the manuscript presented in an intelligible fashion and written in standard English?

Reviewer #1: Yes

Reviewer #2: Yes

Reviewer #3: Yes

5. Review Comments to the Author

Reviewer #1: Overall Impressions

This is an important scientific work that brings out the correlation between the level of fibrinogen and lung functions in old men. From the data and conclusion of the authors, significantly high levels of plasma fibrinogen are observed in the old men with lung problem. Thus the implementation of this parameter as a biomarker or indicator in the diagnosis and/or prognostics of lung dysfunction could be envisaged after a thorough follow up or control of this work.

Corrections

Minor

Page 2, Line 28. ‘percentages of predicted’ should read ‘percentages predicted’

Line 33. Conclusion: ‘We found’ should read ‘It was found’

Line 37. Key words: ‘Respiratory Function Tests’ should read ‘Respiratory Functions’

Page 4, Line 88. ‘plasma is dispensed’ should read ‘plasma was dispensed’

Line 88. ‘C XL reagent is added’ should read ‘C XL reagent was added’

Page 5, Line 103. ‘FEV1 and FVC expressed’ should read ‘FEV1 and FVC were expressed’

Page 7, Line 150. Table 1 ………. ‘ELSA who presented complete data from for fibrinogen’

Major

None detected

Reviewer #2: Thank you for inviting me to review the manuscript entitled “Are elevated plasma fibrinogen associated with lung function? An 8-year follow-up of the ELSA study” It is a good piece of work however, in my opinion it lacks novelty.

Reviewer #3: In this study, the authors reported that a high level of plasma fibrinogen in older English men, but not women, is associated with a decline in lung function. In my opinion, the study is interesting however, I have some major concerns regarding the study design and the inclusion criteria implied in this study. I would like the authors to address the following concerns;

1- The major concern is the inclusion of subjects with pulmonary diseases which affect the lung function in these subjects and thus affect the reliability of the results. These subjects should be excluded to express only the relationship between fibrinogen level and lung function.

2- It is not clear is there any subjects in this study with cardiocirculatiry disorder?

3- The study design regarding the timeline of fibrinogen assessment and Lung function evaluation should be further described briefly. What is meant by wave 2, 4, and 6? Please provide a brief clear description of this part in methodology.

4- Methods: Line 80: “those with a clotting or bleeding disorder or on anticoagulant drugs, those who had ever had…. were not asked to fast.” These cases were stated to be excluded. In this statement they were asked not to fast before sampling? Did you include these cases?

5- In table 2: the significance should be defined with p-values.

6- The limitations: Selection bias should be addressed as a limitation in this study.

6. PLOS authors have the option to publish the peer review history of their article (what does this mean?). If published, this will include your full peer review and any attached files.

Reviewer #1: **Yes: **Dr LUNGA Paul KEILAH

Reviewer #2: No

Reviewer #3: **Yes: **Muhammad Tarek Abdel Ghafar

---

## [Author Response · Author response to Decision Letter 0]

16 Oct 2021

Dear Emiliano Giardina and Reviewers,

We would appreciate the consideration of our Research Article entitled “Are elevated fibrinogen plasma levels associated with lung function? An 8-year follow-up of the ELSA study” for publication in PLOS One. 

Thank you for the reviewers' considerations and suggestions. After corrections we present a new version of the article, and we hope to be in agreement for publication.

Journal Requirements:

We appreciated this comment, and we review the style of the document.

2. In the methods section please provide details regarding how house hold wealth and physical exercise was stratified for analysis.

Thanks for the comment. We added at the references that describle how these variables are stratified. 

Socioeconomic position was based on the National Statistics Socio-Economic Classification (NS-SEC) and quintiles of total wealth, defined from the sum of financial, physical (e.g. businesses, land) and housing wealth, minus debts and pension payments (Banks J, Batty G, Coughlin K, et al (2019) English Longitudinal Study of Ageing: Waves 0-8, 1998-2017. 29th Edition. UK Data Service. SN: 5050, London)

Participants were asked how often they took part in three different types of physical activity: vigorous, moderate and low intensity. The response options were more than once a week, once a week, one to three times a month and hardly ever/never. For the purposes of the analysis, we derived a summary index of physical activity by summing responses to the three physical activity questions after they had been dichotomised around the frequency cut-point of once a week or more often. The derived summary index categorised physical activity as follows: (1) physical inactivity; (2) low-intensity but not vigorous/moderate-intensity physical activity at least once a week; and (3) vigorous/moderate-intensity physical activity at least once a week. (Demakakos, P., Hamer, M., Stamatakis, E. et al. Low-intensity physical activity is associated with reduced risk of incident type 2 diabetes in older adults: evidence from the English Longitudinal Study of Ageing. Diabetologia 53, 1877–1885 (2010). https://doi.org/10.1007/s00125-010-1785-x)

"The English Longitudinal Study of Ageing was developed by a team of researchers based at the University College London, NatCen Social Research, the Institute for Fiscal Studies and the University of Manchester. The NatCen Social Research collected the data. The National Institute of Aging Grant R01AG017644 and a consortium of UK government departments coordinated by the Economic and Social Research Council provide the funding."

"Funders did not participate in the study design, data collection and analysis, decision

to publish or prepare the manuscript."

 Thanks for the corrections, all were executed in the manuscript.

Data Sharing

The English Longitudinal Study of Ageing data are available to the scientific community from the UK Data Service for researchers who meet the criteria for access to confidential data, under conditions of the End User License http://ukdataservice.ac.uk/media/455131/cd137-enduserlicence.pdf. The data can be accessed from: https://beta.ukdataservice.ac.uk/datacatalogue/series/series?id=200011#!/access-data. Contact with the UK Data Service regarding access to the English Longitudinal Study of Ageing can be made through the website https://www.ukdataservice.ac.uk/about-us/contact, by phone +44 (0)1206 872143 or by email at help@ukdataservice.ac.uk. 

Thanks for the comments. The author Ione Schneider has two accounts in PLOS. One is related to username ione.schneider and the other one, to the username ione.jayce. We need to merge these two accounts to be able to validate with Ione’s ORCID (0000-0001-6339-7832). 

The paragraph on ethics statement was removed from the end and is only described in the methods section.

Reviewer #1: 

Corrections

Page 2, Line 28. ‘percentages of predicted’ should read ‘percentages predicted’

Line 33. Conclusion: ‘We found’ should read ‘It was found’

Line 37. Key words: ‘Respiratory Function Tests’ should read ‘Respiratory Functions’

Page 4, Line 88. ‘plasma is dispensed’ should read ‘plasma was dispensed’

Line 88. ‘C XL reagent is added’ should read ‘C XL reagent was added’

Page 5, Line 103. ‘FEV1 and FVC expressed’ should read ‘FEV1 and FVC were expressed’

Page 7, Line 150. Table 1 ………. ‘ELSA who presented complete data from for fibrinogen’

 Thanks for the corrections, all were executed in the manuscript.

Reviewer #2:

Thank you for inviting me to review the manuscript entitled “Are elevated plasma fibrinogen associated with lung function? An 8-year follow-up of the ELSA study” It is a good piece of work however, in my opinion it lacks novelty.

Thanks for the comments. This is the first time that the relation between lung function and fibrinogen is studied in the English population.

Reviewer #3: 

In this study, the authors reported that a high level of plasma fibrinogen in older English men, but not women, is associated with a decline in lung function. In my opinion, the study is interesting however, I have some major concerns regarding the study design and the inclusion criteria implied in this study. I would like the authors to address the following concerns;

1- The major concern is the inclusion of subjects with pulmonary diseases which affect the lung function in these subjects and thus affect the reliability of the results. These subjects should be excluded to express only the relationship between fibrinogen level and lung function.

The questionnaire used contained information about self-reported lung diseases (COPD and asthma). Thus, this self-reported information was used as adjustment variables in the analyses and interactions and multicollinearity between variables were tested, which were not detected.

2- It is not clear is there any subjects in this study with cardiocirculatiry disorder?

The disease high blood pressure, angina, heart attack, congestive heart failure, heart murmur, and abnormal heart rhythm were classified in the presence of one or more. However, they were excluded due to multicollinearity.

3- The study design regarding the timeline of fibrinogen assessment and Lung function evaluation should be further described briefly. What is meant by wave 2, 4, and 6? Please provide a brief clear description of this part in methodology.

Thanks for the comments. We clarified the information about the timeline of fibrinogen and lung function between the waves.

4- Methods: Line 80: “those with a clotting or bleeding disorder or on anticoagulant drugs, those who had ever had…. were not asked to fast.” These cases were stated to be excluded. In this statement they were asked not to fast before sampling? Did you include these cases?

The members with described problems are not eligible to the blood sample. This way, these cases are not included in the analysis. 

5- In table 2: the significance should be defined with p-values.

This information was updated at the table.

6- The limitations: Selection bias should be addressed as a limitation in this study.

Thanks for the comments. We addressed the selection bias to the limitations.

 Yours sincerely

Dr Ione Jayce Ceola Schneider

Affiliate Academic

University College London, Department

---

## [Editor Report · Decision Letter 1]

21 Oct 2021

Are elevated plasma fibrinogen associated with lung function? An 8-year follow-up of the ELSA study

PONE-D-20-39275R1

Dear Dr. Schneider,

We’re pleased to inform you that your manuscript has been judged scientifically suitable for publication and will be formally accepted for publication once it meets all outstanding technical requirements.

Kind regards,

Emiliano Giardina

Academic Editor

PLOS ONE

---

## [Editor Report · Acceptance letter]

26 Oct 2021

PONE-D-20-39275R1 

Are elevated plasma fibrinogen associated with lung function? An 8-year follow-up of the ELSA study 

Dear Dr. Schneider:

I'm pleased to inform you that your manuscript has been deemed suitable for publication in PLOS ONE. Congratulations! Your manuscript is now with our production department. 

Kind regards, 

on behalf of

Dr. Emiliano Giardina 

Academic Editor

PLOS ONE